# NiCo Prussian-Blue-Derived Cobalt–Nickel-Layered Double Hydroxide with High Electrochemical Performance for Supercapacitor Electrodes

Qihao Yin [1], Bo Gao [1,*], Haiyang Fu [1] and Liang Hu [2]

1   School of Metallurgy, Northeastern University, Shenyang 110819, China
2   School of Materials Science and Engineering, Shenyang Ligong University, Shenyang 110159, China
*   Correspondence: gaob@smm.neu.edu.cn

**Abstract:** High-performance electrode materials are crucial to the improvement of the supercapacitor performance index. $Ni_2Co_1HCF@CoNi$-LDH composites with a core–shell structure were prepared by a combination of coprecipitation and constant potential electrodeposition, and the microscopic morphology and phase composition of the composites were characterized by XRD, SEM, FTIR and XPS. The results showed that the NiCo Prussian blue ($Ni_2Co_1HCF$) was grown on the nickel foam (NF) substrate by in situ etching, while the nickel–cobalt double hydroxide (CoNi-LDH) was covered on the NiCo Prussian blue surface by electrodeposition, and the composite still retained the cubic skeleton morphology of the NiCo Prussian blue. The electrochemical properties of the composites were investigated using a three-electrode system in 2 M KOH. The results showed that their discharge specific capacity was as high as 1937 $F \cdot g^{-1}$ at a current density of 1 $A \cdot g^{-1}$ and still had 81.3% capacity retention at 10 $A \cdot g^{-1}$, and they exhibited an excellent rate capability. The capacity retention rate was 87.1% after 1000 cycles at 5 $A \cdot g^{-1}$ and, thus, the composite material has good application prospects as a supercapacitor electrode material.

**Keywords:** coprecipitation; constant potential electrodeposition; cubic skeleton; supercapacitor; electrode material





## 1. Introduction

With the increasingly serious environmental pollution and energy crisis caused by the burning of fossil fuels, the development of sustainable and environmentally friendly energy storage equipment has become an urgent problem to be solved. As a new type of energy storage device, supercapacitors have received much attention because of their high power density, fast charging and discharging rates, long cycle life and sufficient safety [1]. Generally, supercapacitors can be classified into three different types, including electric double-layer capacitors (EDLCs), pseudocapacitors and asymmetric supercapacitors [2]. The charge storage mechanism of pseudocapacitor electrode materials involves the Faraday process of electrochemical redox reactions and, thus, possesses a higher specific capacitance [3]. Therefore, the synthesis and modification of new types of pseudocapacitor electrode materials have become the focus of current research.

Prussian blue analogues (PBAs), as a kind of metal–organic framework (MOF), have attracted much attention because of their excellent porosity, large specific surface area, and adjustable structure [4]. The chemical formula for PBAs is $A_xM_A[M_B(CN)_6]_y \cdot zH_2O$, where A is an alkali metal cation, such as $K^+$ or $Na^+$. $M_A$ and $M_B$ are transition metal cations, such as $Co^{2+}$, $Ni^{2+}$, and $Fe^{3+}$, with a unique face-centered cubic structure, which is a typical pseudocapacitive electrode material [5]. In addition, PBAs have an open three-dimensional skeleton, which not only allows for the rapid intercalation/extraction of $K^+$ and $Na^+$ but also withstands the structural stress changes caused by the doping process of alkali metal ions and, thus, has a better cycling stability [6]. However, the poor conductivity and lower

specific capacity limits the further application of PBAs in supercapacitors. In order to solve this problem, scholars have conducted extensive research in recent years. Shen et al. obtained $(Ni, Co)Se_2$ by vacuum heating selenization using $Ni_3[Co(CN)_6]_2$ synthesized by coprecipitation as a precursor, and the material reached a specific capacity of 778 F·g$^{-1}$ at a current density of 1 A·g$^{-1}$ [7]. Xiong et al. grew $NiCo_xFe_{1-x}$-PBA directly on nickel foam by in situ etching, and the material exhibited an excellent rate capability with a high specific capacitance of 2242 mF·cm$^{-2}$ at a current density of 1 mA·cm$^{-2}$ and a capacity retention of 43.1%, even at a high current density of 20 mA·cm$^{-2}$ [8].

In this work, we propose a low energy consumption and relatively simple two-step method to synthesize $Ni_2Co_1$HCF@CoNi-LDH/NF composites. NiCo Prussian blue was synthesized by the coprecipitation method, which was grown in situ on the surface of a nickel foam substrate, and then a layer of cobalt–nickel double hydroxide was coated on the surface of the NiCo Prussian blue by the constant potential electrodeposition method. The phase composition and morphology were analyzed by XRD, SEM, FTIR and XPS, and the composite exhibited a regular cubic framework morphology. The electrochemical performance was tested in a three-electrode system, and the specific capacitance of the $Ni_2Co_1$HCF@CoNi-LDH/NF composite tremendously improved to 1937 F·g$^{-1}$ at a current density of 1 A·g$^{-1}$; even at a high current density of 10 A·g$^{-1}$, the capacity retention rate could still reach 81.3%, showing excellent electrochemical performance. This study provides a new idea for the design of pseudocapacitive electrode materials.

## 2. Experimental Section

### 2.1. Materials

Nickel nitrate hexahydrate $(Ni(NO_3)_2 \cdot 6H_2O)$, cobalt nitrate hexahydrate $(Co(NO_3)_2 \cdot 6H_2O)$ and ethanol absolute and potassium hydroxide (KOH) were purchased from Sinopharm Chemical Reagent Co., Ltd., Shanghai, China. Trisodium citrate dihydrate $(N_6H_5Na_3O_7 \cdot 2H_2O)$ and potassium ferricyanide $(K_3Fe(CN)_6)$ were purchased from Aladdin, Shanghai, China. All reagents were of analytical grade without further purification. Nickel foam with a thickness of 1.7 mm was used as a fluid collector. The surface oxide layer should be removed before use during the pretreatment. The nickel foam was dipped into acetone and 3 M of diluted hydrochloric acid and ultrasonically cleaned, followed by several washes with ethanol absolute and deionized water.

### 2.2. Synthesis of $Ni_2Co_1$HCF/NF

$Ni_2Co_1$HCF/NF was prepared using the coprecipitation method under a room temperature environment. The specific experimental steps were as follows: First, a total of 30 mM $Ni(NO_3)_2 \cdot 6H_2O$ and $Co(NO_3)_2 \cdot 6H_2O$ was fully dissolved in deionized water at a molar ratio of 2:1. Subsequently, sodium citrate was added and sonicated to form homogeneous solution A. Then, 20 mM of $K_3Fe(CN)_6$ was fully dissolved in deionized water by magnetic stirring to form solution B. Finally, solution B was gradually added dropwise to solution A, and the two solutions were mixed thoroughly by vigorous stirring. The treated nickel foam $(1 \times 1 \text{ cm}^2)$ was immersed in the mixed solution and aged for 12 h. After the reaction, the nickel foam was removed and washed several times with deionized water and ethanol absolute and dried overnight at 60 °C in a vacuum drying oven.

### 2.3. Synthesis of $Ni_2Co_1$HCF@CoNi-LDH/NF

A total of 10 mM $Ni(NO_3)_2 \cdot 6H_2O$ and $Co(NO_3)_2 \cdot 6H_2O$ was dissolved at a certain ratio in deionized water to obtain a clarified solution, followed by constant potential electrodeposition using a three-electrode system. The dried $Ni_2Co_1$HCF/NF was used as the working electrode, a platinum sheet as the counter electrode, saturated calomel electrode (SCE) as the reference electrode and the prepared mixed solution as the electrolyte to form a three-electrode system, followed by a constant potential of −1 V applied to the system and a deposition time of 600 s. After the electrodeposition, the nickel foam deposited with active material was taken out, rinsed several times with deionized water

and anhydrous ethanol and dried in a vacuum drying oven at 60 °C for 6 h. For comparison, we repeated the above electrodeposition process using the pretreated bare nickel foam as the working electrode to obtain CoNi-LDH/NF.

*2.4. Characterizations*

The samples were analyzed by XRD using an X'Pert PRO-DY2198 diffractometer (scan range: 10°~90°, sweep speed: 3°/min, Panalytical, Malvern, UK) to determine their phase composition. The morphology and size of the material were observed using a Zeiss Gemini 300 field emission scanning electron microscope (SEM, Zeiss, Jena, Germany) with energy-dispersive X-ray spectrometry (EDS) mapping. The Fourier transform infrared (FTIR) spectra of the samples were recorded using a Bruker Vertex 80v spectrometer (scan range: 400 to 4000 cm$^{-1}$, Bruker, Billerica, MA, USA) to determine the functional groups and molecular structures contained in the samples. X-ray photoelectron spectroscopy (XPS) was conducted using a Kratos Axis Ultra-DLD photoelectron spectrometer (Kratos Analytical Ltd, Manchester, UK) with an Al K$\alpha$ X-ray excitation source to determine the elemental composition and valence states of the sample surface.

*2.5. Electrochemical Measurements*

Measurements of the electrochemical performance at room temperature were carried out in a CHI760E electrochemical working station, Shanghai, China. The three-electrode system was used as a test device in which a platinum sheet was used as the counter electrode, a saturated calomel electrode (SCE) as the reference electrode and 2 M KOH as the electrolyte. The working electrode was nickel foam with active substance, and no binder was used. Cyclic voltammetry (CV) tests were performed in a voltage range of 0–0.5 V. In this study, a low sweep second rate (1–2 mV·s$^{-1}$) was used to investigate the pseudocapacitance behavior of the electrodes. Electrochemical impedance spectroscopy (EIS) was carried out in a frequency range of 0.01 Hz to 100 kHz, with an amplitude of 5 mV. Galvanostatic charge/discharge (GCD) tests were conducted at current densities of 1, 2, 5 and 10 A·g$^{-1}$, respectively, with voltage windows ranging from 0 to 0.4 V. The specific capacitance of the electrode material is usually calculated according to Equation (1) [9]:

$$C = \frac{2i_m \int V dt}{V^2 \big|_{V_i}^{V_f}} \tag{1}$$

where C is the specific capacitance (F·g$^{-1}$), $i_m$ is the current density (A·g$^{-1}$), $\int V dt$ is the integrated area of the GCD curve during the discharge process, $V_i$ (V initial) is the initial potential value during the discharge process, and $V_f$ (V final) is the final potential value during the discharge process.

## 3. Results and Discussion

*3.1. Morphology and Structure Characterization*

We synthesized Ni-Co Prussian blue by coprecipitation (Figure 1a), the basic principle of which is to use transition metal cations to replace high spin Fe (HS-Fe) in ferricyanate to reduce crystal defects and decrease vacancy water, thus improving the electrochemical performance of the electrode materials. The doping of Ni$^{2+}$ mitigates the cell volume and stress changes caused during K$^+$ migration and improves the material stability, while the doping of Co$^{2+}$ is used to enhance the electrochemical activity of the material to improve its discharge capacity [10]. The purpose of adding sodium citrate is to control the Prussian blue morphology and size by reducing the nucleation rate of crystals through its chelating effect [11]. In order to obtain supercapacitor electrode materials with high capacity, good multiplicative performance and high cycling stability, we subsequently grew CoNi-LDH on the Ni$_2$Co$_1$HCF/NF surface using constant potential electrodeposition,

and the electrochemical reactions occurring during the electrodeposition process can be expressed as [12]:

$$NO_3^- + 7H_2O + 8e^- \rightarrow NH_4^+ + 10OH^- \tag{2}$$

$$Ni^{2+} + 2OH^- \rightarrow Ni(OH)_2 \tag{3}$$

$$Co^{2+} + 2OH^- \rightarrow Co(OH)_2 \tag{4}$$

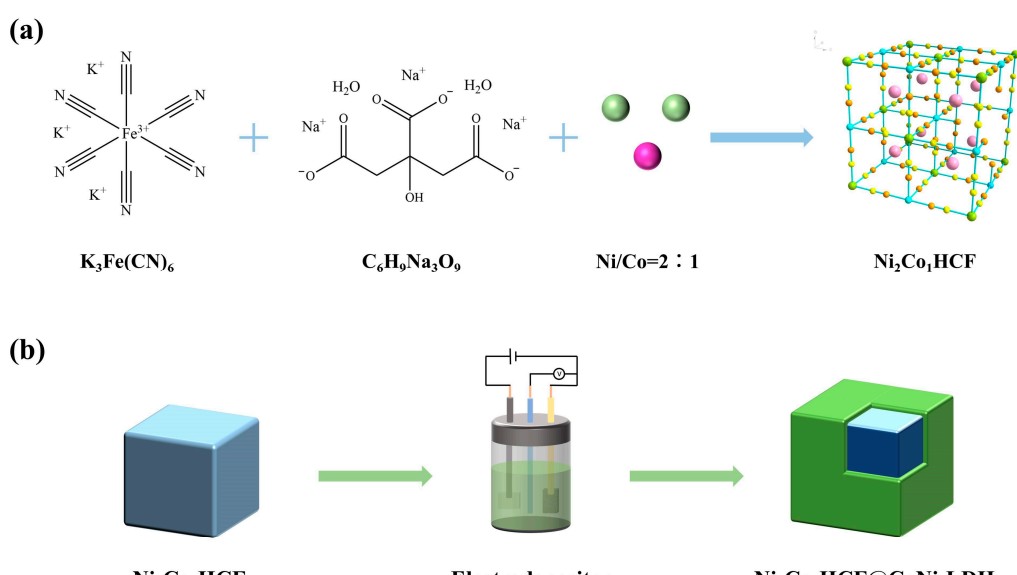

**Figure 1.** Schematic illustration of synthesizing (**a**) $Ni_2Co_1HCF$; (**b**) $Ni_2Co_1HCF@CoNi-LDH/NF$.

Figure 2a shows the XRD pattern of $Ni_2Co_1HCF$. It can be seen that this XRD pattern shows a distinct sharp peak, indicating that the sample has good crystallinity [13]. In addition, the diffraction peaks at 17.3°, 24.6° 34.85°, 39.4° and 43.4° correspond to the (200), (220), (400), (420) and (422) crystal planes of the NiHCF (JCPDS No. 46-0906) and CoHCF (JCPDS No. 86-0502) standard cards, with a typical face-centered cubic structure, which proves that the $Ni_2Co_1HCF$ samples were successfully synthesized. Figure 2b shows the XRD patterns of the nickel–cobalt-layered bimetallic hydroxide. It can be seen that the diffraction peaks at 11.3°, 22.7°, 34.4° and 59.9° correspond to the (003), (006), (012) and (110) crystal planes of the $Ni(OH)_2$ (JCPDS No.38-0715) standard card, while the diffraction peaks at 11.5°, 23.2°, 34.4° and 59.8° correspond to the (003), (006), (102) and (110) crystal planes of the $Co_5 (O_{9.48}H_{8.52}) NO_3$ (JCPDS No.38-0715) standard card, which proves that the CoNi-LDH was successfully synthesized. Figure 2c shows the comparative XRD patterns of the $Ni_2Co_1HCF$, CoNi-LDH/NF and $Ni_2Co_1HCF@CoNi$-LDH composites. It can be seen that the diffraction peaks at 15.12°, 17.3°, 24.6° and 34.85° correspond to the characteristic peaks of $Ni_2Co_1HCF$, while the diffraction peaks at 11.5°, 34.4°, 59.8° and 69.5° correspond to the characteristic peaks of CoNi-LDH, indicating that the two substances were successfully compounded.

In order to further investigate the composition of the functional groups and the bonding mode in the composites, Fourier transform infrared spectroscopy (FTIR) analysis was performed on the samples. Figure 2d shows the FTIR comparison of $Ni_2Co_1HCF$ and $Ni_2Co_1HCF@CoNi-LDH$. The absorption peaks at 1617 and 3552 cm$^{-1}$ are associated with the stretching of water and hydroxyl (-OH), indicating the presence of crystalline water in the Prussian blue framework [14]. The absorption peaks at 595 and 2097 cm$^{-1}$ can be attributed to the characteristic stretching of C≡N [15]. Precisely, the two characteristic peaks at 2097 and 2161 cm$^{-1}$ correspond to the stretching vibration of $Co^{2+}$-C≡N-$Ni^{3+}$ and $Co^{3+}$-C≡N-$Ni^{2+}$, respectively, which indicate that the free $Ni^{2+}/Ni^{3+}$ and $Co^{2+}/Co^{3+}$ in the electrolyte react chemically with $[Fe(CN)_6]^{3-}$ in $Ni_2Co_1HCF$. The absorption peak at

1384 cm$^{-1}$ is attributed to the vibration of the N-O bond in NO$_3^-$, and the absorption peak located at 800 to 400 cm$^{-1}$ is associated with the vibration of M-O and M-OH (M for Ni and Co) [16]. The Ni$_2$Co$_1$HCF and Ni$_2$Co$_1$HCF@CoNi-LDH were successfully compounded, as shown by comparing the characteristic peaks and chemical bonding species in the FTIR spectra.

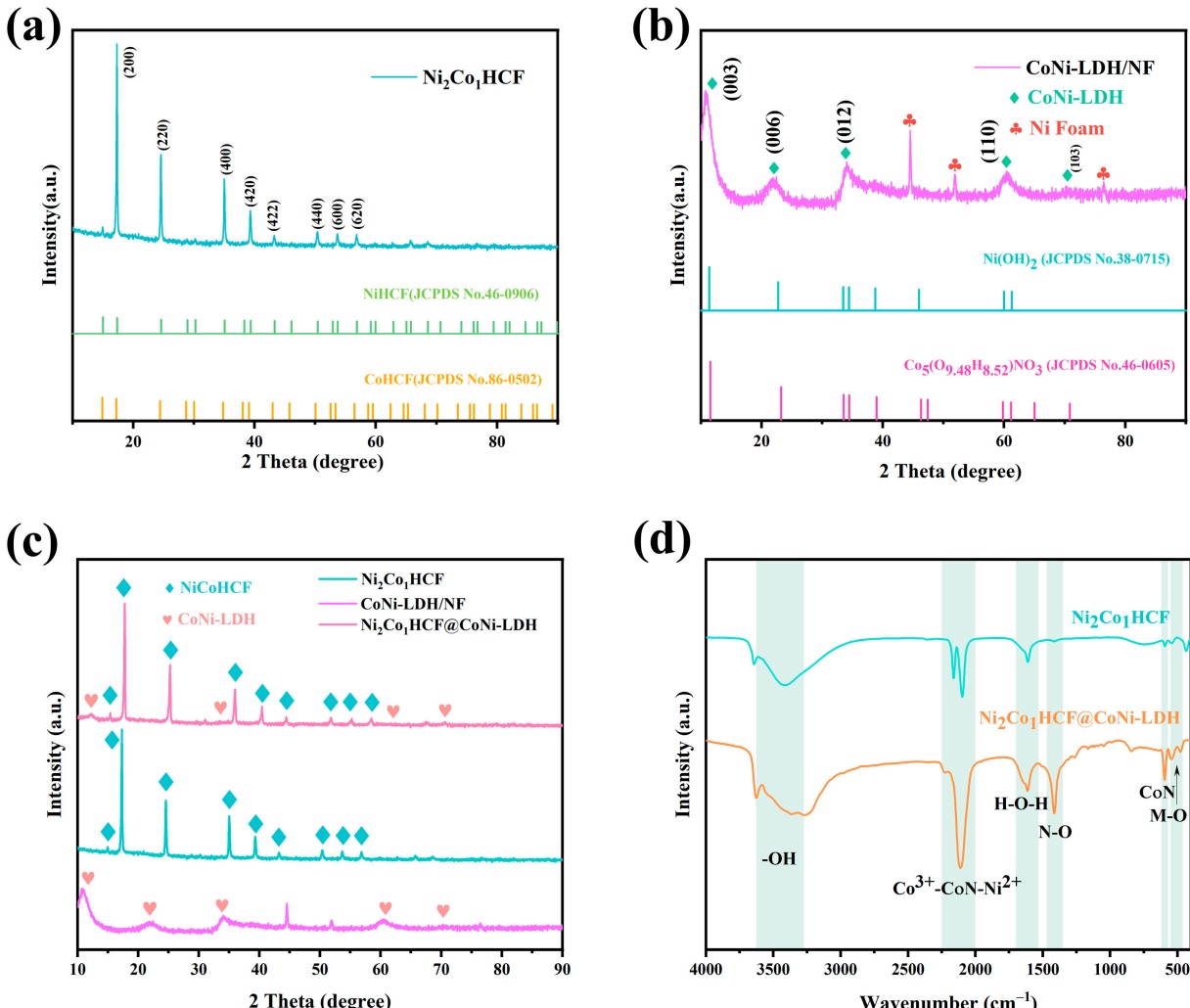

**Figure 2.** XRD patterns of (**a**) Ni$_2$Co$_1$HCF; (**b**) CoNi-LDH/NF; (**c**) Ni$_2$Co$_1$HCF@CoNi-LDH composites. (**d**) Fourier transform infrared spectra of the Ni$_2$Co$_1$HCF and Ni$_2$Co$_1$HCF@CoNi-LDH composites.

Figure 3a,b show SEM images of the NiCo Prussian blue samples. It can be seen that the Ni$_2$Co$_1$HCF/NF synthesized by the coprecipitation method has a regular cubic skeleton morphology. Notably, the chelating effect of sodium citrate not only slowed down the nucleation rate of the crystals but also contributed to the formation of the cubic skeleton. Figure 3c shows the microscopic morphology of the Ni$_2$Co$_1$HCF@CoNi-LDH/NF composites. The surface of the cube skeleton profile is a typical morphology of cobalt–nickel-layered bimetallic hydroxide, which is a three-dimensional nanoarray structure with several nanosheets crossing each other. Obviously, CoNi-LDH was successfully coated on surface of the NiCo Prussian blue samples, both successful compounds. In addition, the Ni$_2$Co$_1$HCF@CoNi-LDH/NF composite synthesized using the two-step method was uniformly distributed on the surface of the nickel foam (Figure 3d). From the EDS mapping of the composite (Figure 3e), it can be found that the Ni, Co, Fe, C, N and O elements were uniformly distributed in the material, which to some extent indicates the high purity and

crystal integrity of the particles [17]. In addition, all these elements are constituents of NiCo Prussian blue with CoNi-LDH, indicating the successful synthesis of the $Ni_2Co_1HCF@CoNi$-LDH/NF composites.

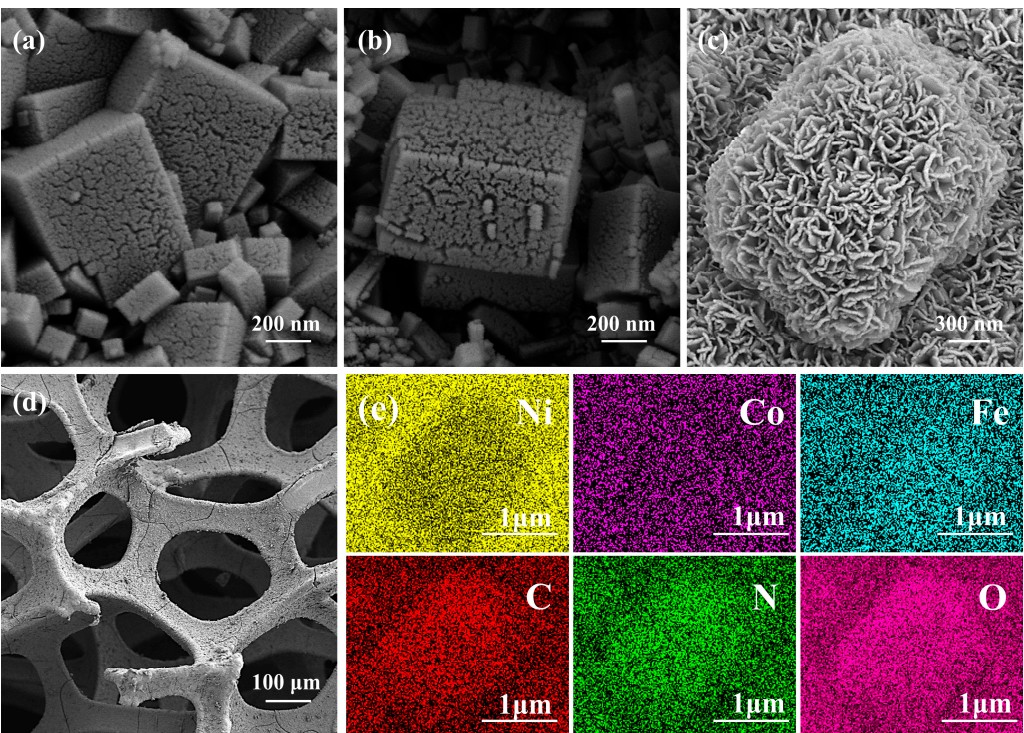

**Figure 3.** SEM images of (**a**), (**b**) $Ni_2Co_1HCF/NF$, (**c**) $Ni_2Co_1HCF@CoNi$-LDH/NF; (**d**) distribution of the composites on nickel foam; (**e**) EDS mapping of $Ni_2Co_1HCF@CoNi$-LDH/NF.

In order to further investigate the chemical composition, electronic structure and valence of the material surface, we performed XPS analysis on the $Ni_2Co_1HCF/NF$ and $Ni_2Co_1HCF@CoNi$-LDH/NF composites. The elements contained in the XPS survey spectrum (Figure 4a) are all constituent elements of the two materials, proving that the Prussian blue composite was successfully prepared. Figure 4b compares the high-resolution Co 2p spectra of the two materials. The Gaussian fitting of the Co 2p spectrum obtained for the $Ni_2Co_1HCF/NF$ shows two strong separation peaks at 780.7 and 796.1 eV, assigned to spin orbitals of $Co^{2+}$ $2p_{3/2}$ and $Co^{2+}$ $2p_{1/2}$, respectively, and the peaks at 783.6 and 797.1 eV can be attributed to the spin orbitals of $Co^{3+}$ $2p_{3/2}$ and $Co^{3+}$ $2p_{1/2}$, respectively [18]. Similarly, the separation peaks at 855.4/857.4 and 872.6/873.8 in the high-resolution Ni 2p spectrum (Figure 4c) are assigned to the $2p_{3/2}$ and $2p_{1/2}$ spin orbitals of $Ni^{2+}/Ni^{3+}$, respectively [19]. It is noteworthy that the binding energy of Co and Ni in the $Ni_2Co_1HCF@CoNi$-LDH/NF composites decreased by 0.3–0.6 eV compared to $Ni_2Co_1HCF/NF$, implying a charge transfer between Fe, Co and Ni atoms [20]. In addition, the peak intensities of Co and Ni in the composites increased in the XPS patterns, which indicates that CoNi-LDH grew on the NiCo-based Prussian blue surface and that CoNi-LDH successfully bonded with $Ni_2Co_1HCF$. The Gaussian fitting of the N1s spectra of both materials (Figure 4d) reveal that the separated peaks of $Ni_2Co_1HCF/NF$ at 398.1, 399.1 and 401.3 eV were attributed to pyridine nitrogen, pyrrole nitrogen and quaternary nitrogen, respectively [21]. Compared with $Ni_2Co_1HCF/NF$, the N1s pattern of the composite showed two separate peaks at 403 and 406.4 eV, which can be attributed to the N-H and N-$O_x$ in the material. The above analysis is consistent with the FTIR results.

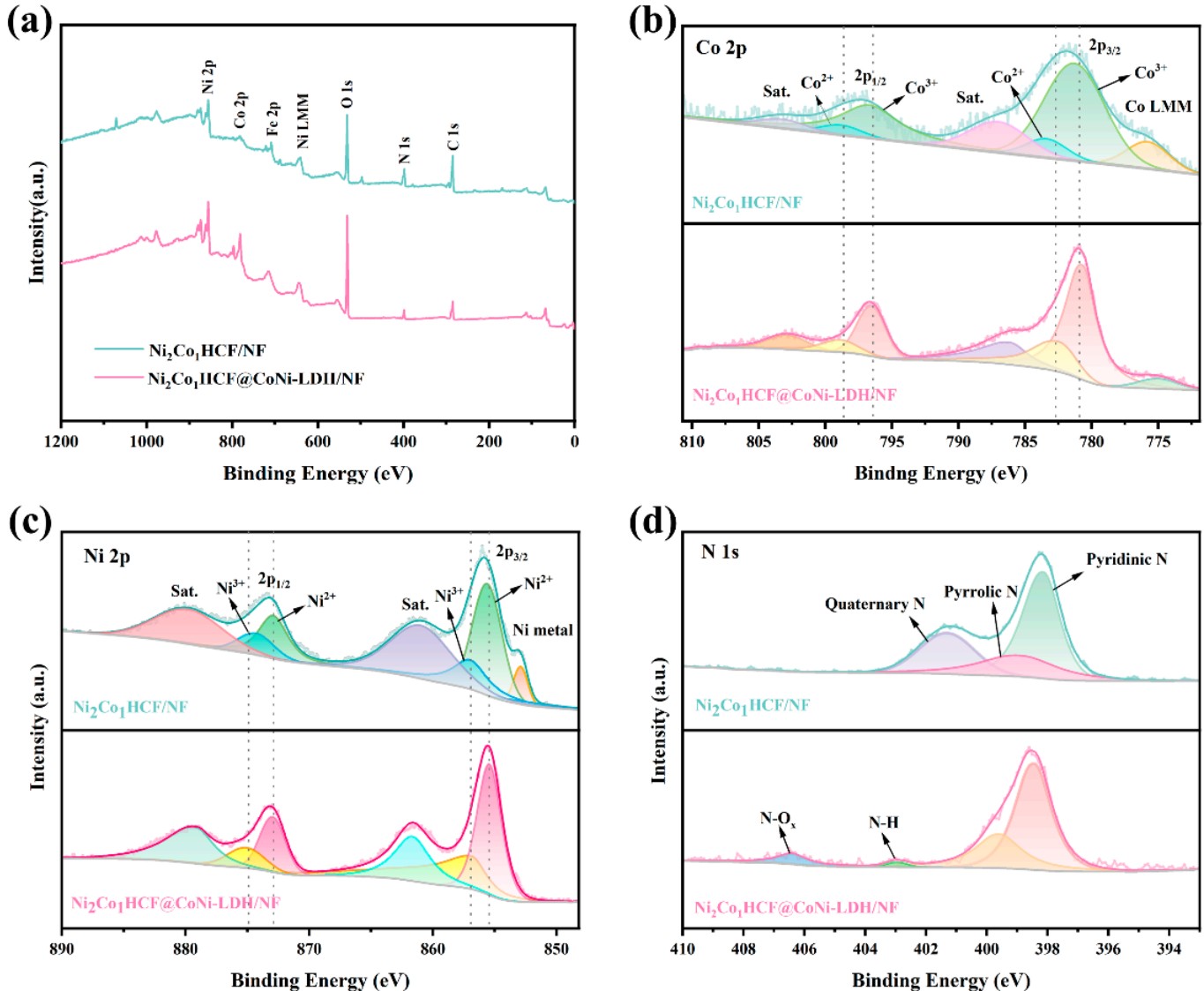

**Figure 4.** XPS spectra of (**a**) survey spectrum; (**b**) Co 2p peak; (**c**) Ni 2p peak; (**d**) N 1s peak of the Ni$_2$Co$_1$HCF/NF and Ni$_2$Co$_1$HCF@CoNi-LDH/NF composites.

### 3.2. Electrochemical Performance

To further evaluate the supercapacitance performance of the electrode materials, electrochemical tests were carried out on the three materials in 2 M KOH aqueous solution. The GCD curves of the Ni$_2$Co$_1$HCF@CoNi-LDH/NF composites at different current densities are shown in Figure 5a. According to Equation (1), the material shows a high discharge specific capacity of 1937 F·g$^{-1}$ at 1 A·g$^{-1}$; even at a higher current density of 10 A·g$^{-1}$, a remarkable specific capacitance of 1575 F·g$^{-1}$ can still be retained demonstrating an efficient electron conduction rate. In addition, different from the linear charge–discharge process of the double-layer capacitive behavior, both the charging and discharging processes of the composite showed an obvious platform, indicating that the electrode material is a pseudocapacitive energy storage mechanism. This means that the Faraday reaction occurs during the charging and discharging process, and the reaction equation can be expressed as [22,23]:

$$M^{II}\left[Fe^{III}(CN)_6\right] + K^+ + e^- \leftrightarrow KM^{II}\left[Fe^{II}(CN)_6\right](M = Co, Ni) \tag{5}$$

$$Co(OH)_2 + OH^- \leftrightarrow CoOOH + H_2O + e^- \tag{6}$$

$$CoOOH + OH^- \leftrightarrow CoO_2 + H_2O + e^- \qquad (7)$$

$$Ni(OH)_2 + OH^- \leftrightarrow NiOOH + H_2O + e^- \qquad (8)$$

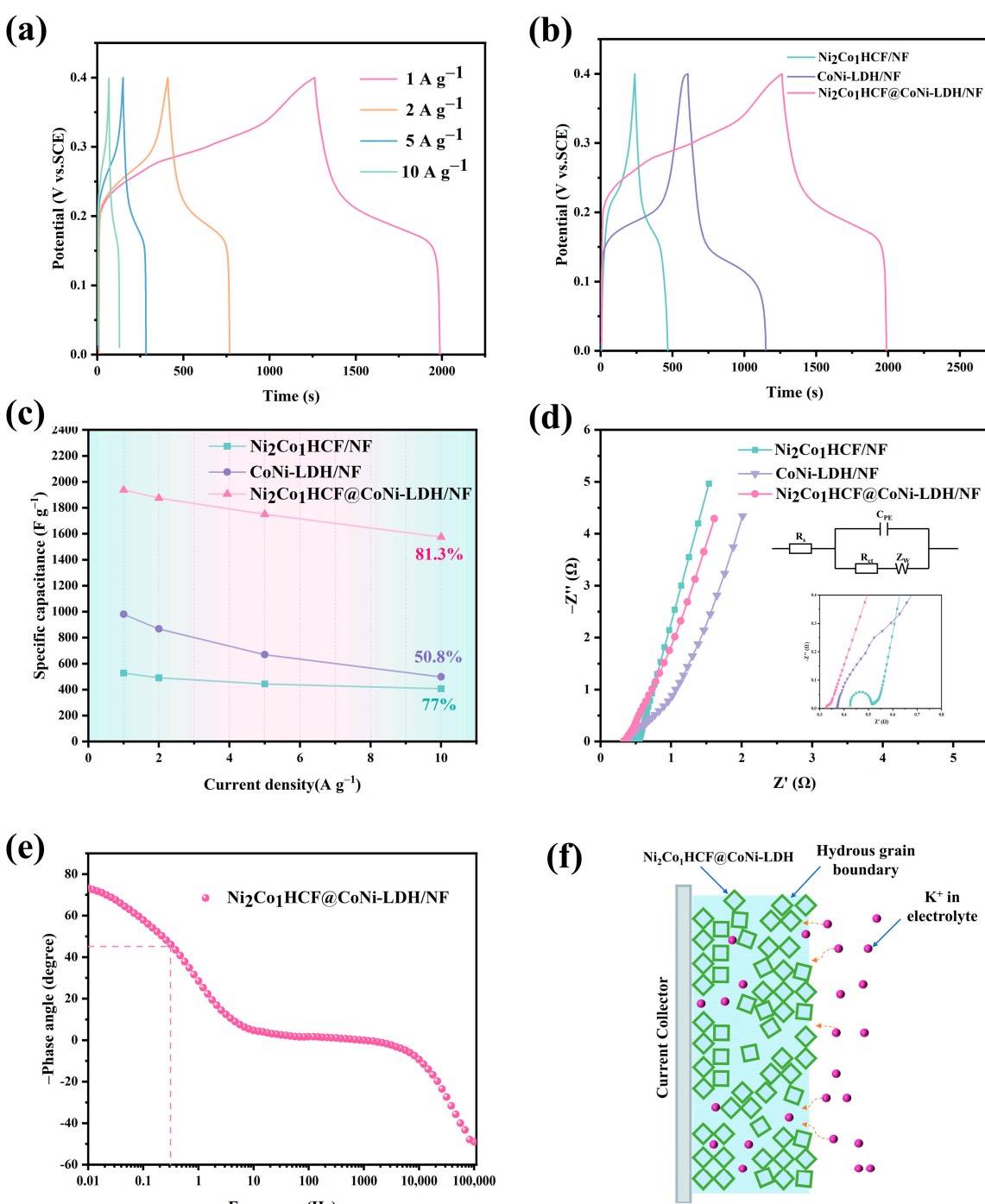

**Figure 5.** (**a**) GCD curves of the $Ni_2Co_1HCF@CoNi-LDH/NF$ composites at different current densities; (**b**) comparison of the GCD curves at a current density of 1 A·$g^{-1}$; (**c**) comparison of the specific capacitance variations at different current densities of the materials; (**d**) comparison of the Nyquist plots; (**e**) Bode diagram of the composite materials; (**f**) schematic diagram of the electrode material diffusion process.

Compared with the two materials before the composite, $Ni_2Co_1HCF@CoNi-LDH/NF$ has a longer discharge time within the same voltage window, indicating that its specific capacitance is the largest. From Equation (1), it can be calculated that the specific capacitances

corresponding to $Ni_2Co_1HCF/NF$ and $CoNi$-$LDH/NF$ were 527.5 and 980.25 $F\cdot g^{-1}$. The line graph of the variation of the specific capacity with the current density is calculated from the GCD curves at different current densities, as shown in Figure 5c. It can be intuitively seen that the specific capacity decreased with the increase in the current density. Moreover, $CoNi$-$LDH/NF$ had the worst capacity retention at 10 $A\cdot g^{-1}$ with only 50.8%, which is attributed to its poor electrical conductivity. However, after compounding $CoNi$-$LDH$ with $NiCo$ Prussian blue, an MOF material with better conductivity, the rate performance of the composite significantly improved, with a capacity retention of 81.3% at 10 $A\cdot g^{-1}$. Based on the above analysis, we believe that there is a synergistic effect between $CoNi$-$LDH$ and $NiCo$ Prussian blue, which means that $CoNi$-$LDH$ enhances the specific capacity and $NiCo$ Prussian blue improves the charge transfer process. Figure 5d shows the AC impedance spectrum results for the three materials, and the circuit description code of $R_s(C(R_{ct}W))$ was used for the fitted circuit. The high-frequency region in the figure is a semicircle, which was controlled by the charge transfer process at this time, while the low-frequency region was the straight line of a linear relationship, which is controlled by the semi-infinite diffusion step [24]. The intersection of the spectral line with the real axis and the diameter of the arc represent the series resistance ($R_s$) and the charge transfer resistance ($R_{ct}$) of the electrodes, respectively, while the slanted part represents the Warburg impedance ($Z_w$) caused by semi-infinite diffusion. The $R_s$ is the series value of the solution resistance and the internal resistance of the active material, while the semi-infinite diffusion process refers to the diffusion region in the solution, i.e., the region where the concentration gradient of the diffusing particles is a certain value at a constant state and the thickness of the diffusion layer is infinite. A schematic diagram of the electrode material diffusion process is shown in Figure 5f. Comparing the results of the charge transfer impedance ($R_{ct}$) shows that $Ni_2Co_1HCF@CoNi$-$LDH/NF$ (0.02 $\Omega$) < $Ni_2Co_1HCF/NF$ (0.092 $\Omega$) < $CoNi$-$LDH/NF$ (0.4 $\Omega$), which indicates that the composites have better electrical conductivity, which is consistent with the results expressed by the ratio curves. In addition, the series resistance $R_s$ of the three materials were all small (approximately 0.37 $\Omega$), which is due to the active substance growing directly on the surface of the nickel foam instead of being attached by binder. Figure 5e shows a Bode diagram of the composite; when the phase angle was $-45°$, the time constant $\tau 0$ ($\tau_0 = 1/f_0$) of the relaxation process could be obtained as 3.125 s based on its response frequency $f_0$ (0.32 Hz), indicating that at frequencies below this frequency, most of the energy storage form is pure capacitive behavior [25]. The results of the fitted values for each component in the equivalent circuit diagram are shown in Table 1.

**Table 1.** Parameter values of the different samples obtained by the equivalent circuit simulation.

| Samples | $R_s/\Omega$ | C/F | $R_{ct}/\Omega$ | $Z_w/\Omega$ |
|---|---|---|---|---|
| $Ni_2Co_1HCF/NF$ | 0.427 | 0.045 | 0.092 | 0.112 |
| $CoNi$-$LDH/NF$ | 0.374 | 0.106 | 0.4 | 0.181 |
| $Ni_2Co_1HCF@CoNi$-$LDH/NF$ | 0.329 | 0.0934 | 0.02 | 0.152 |

Figure 6a shows the comparative CV curves of $Ni_2Co_1HCF/NF$ and $Ni_2Co_1HCF@CoNi$-$LDH/NF$ at 1 $mV\cdot s^{-1}$. It can be clearly seen that the area enclosed by the CV curve of the $Ni_2Co_1HCF@CoNi$-$LDH/NF$ composite was significantly higher than that of the NiCo-based Prussian blue, indicating that the composite possessed a higher specific capacitance. The evident redox couples in the CV curves can be observed, manifesting the presence of Faradaic reactions of the electrodes [26]. It is worth noting that the current difference between the oxidation and reduction peaks of the composite was smaller than that of NiCo Prussian blue, indicating an improvement in its reversibility. From the CV curves of the composites at different scan rates (Figure 6b), it can be seen that the redox peaks gradually shifted with the increase in the scan rate, indicating the polarization of the electrode materials. In order to better analyze the charge storage mechanism and diffusion kinetics of the composites, we performed relevant calculations using the CV curves of the composites

with low scan rates. Theoretically, the relationship between the voltammetric current and scan rate can be expressed as [27]:

$$i = av^b \qquad (9)$$

$$\log i = b \log v + \log a \qquad (10)$$

where Equation (10) is transformed from Equation (9), i is the peak current, v is the scan rate, and a and b are constants. If the value of b is 0.5, the redox is a semi-infinite diffusion process, and a b value of 1 indicates a totally pseudocapacitor-controlled capacitive process [28]. We performed a linear fit to the discrete points using Equation (10) with logv as the *x*-axis and logi as the *y*-axis, and the results are shown in Figure 6c. The fitted b values of 0.71 and 0.64 for the anodic and cathodic peaks, respectively, indicate that the charge storage of the composite was controlled by the ion diffusion process in conjunction with the pseudocapacitive behavior. To further determine which process dominates the charge storage in the composite, we performed calculations using Equations (11) and (12) [29]:

$$i(V) = k_1 v + k_2 v^{1/2} \qquad (11)$$

$$\frac{i}{v^{1/2}} = k_1 v^{1/2} + k_2 \qquad (12)$$

where Equation (12) is obtained after separating the constants of Equation (11), i is the current at a fixed voltage, v is the sweep speed, and $k_1$ and $k_2$ are constants. The pseudocapacitance contribution of the composite at a scan rate of 1 mV·s$^{-1}$ is shown in Figure 6d, and it can be seen that the pseudocapacitance contribution is 61%, indicating that the pseudocapacitance dominates the charge storage at this time. Figure 6e shows the proportion of the pseudocapacitance contribution at different scanning rates; it can clearly be seen that the proportion of the diffusion process decreased with the increase in the scanning rate. Finally, the composite was tested for cycling stability, and it still had 87.1% capacity retention after 1000 cycles at a current density of 5 A·g$^{-1}$. A comparison of the specific capacity and rate capability of several types of materials is shown in the Table 2, which shows that the composite material synthesized in our research has a higher specific capacity and better multiplicative performance, indicating that the composite material is an ideal electrode material for supercapacitors and has good prospects for application in the field of electrochemical energy storage.

**Table 2.** Comparison of the specific capacities and rate capabilities of several types of materials.

| Materials | Electrolyte | Specific Capacity (F·g$^{-1}$) | Capacitance Retention (Ratio) | Reference |
|---|---|---|---|---|
| Ni$_2$CoHCF/NF | 2M KOH | 1300 (0.5 A·g$^{-1}$) | 53% | [30] |
| CoHCC | 3M KOH | 1720 (1 A·g$^{-1}$) | 61.2% | [31] |
| Ni$_4$Co$_1$-LDH | 2M KOH | 1620 (1 A·g$^{-1}$) | 68% | [26] |
| Ni$_2$Co$_1$-LDH | 2M KOH | 963 (0.5 A·g$^{-1}$) | 76.8% | [32] |
| Ni$_2$Co$_1$HCF@CoNi-LDH/NF | 2M KOH | 1937 (1 A·g$^{-1}$) | 81.3% | This Work |

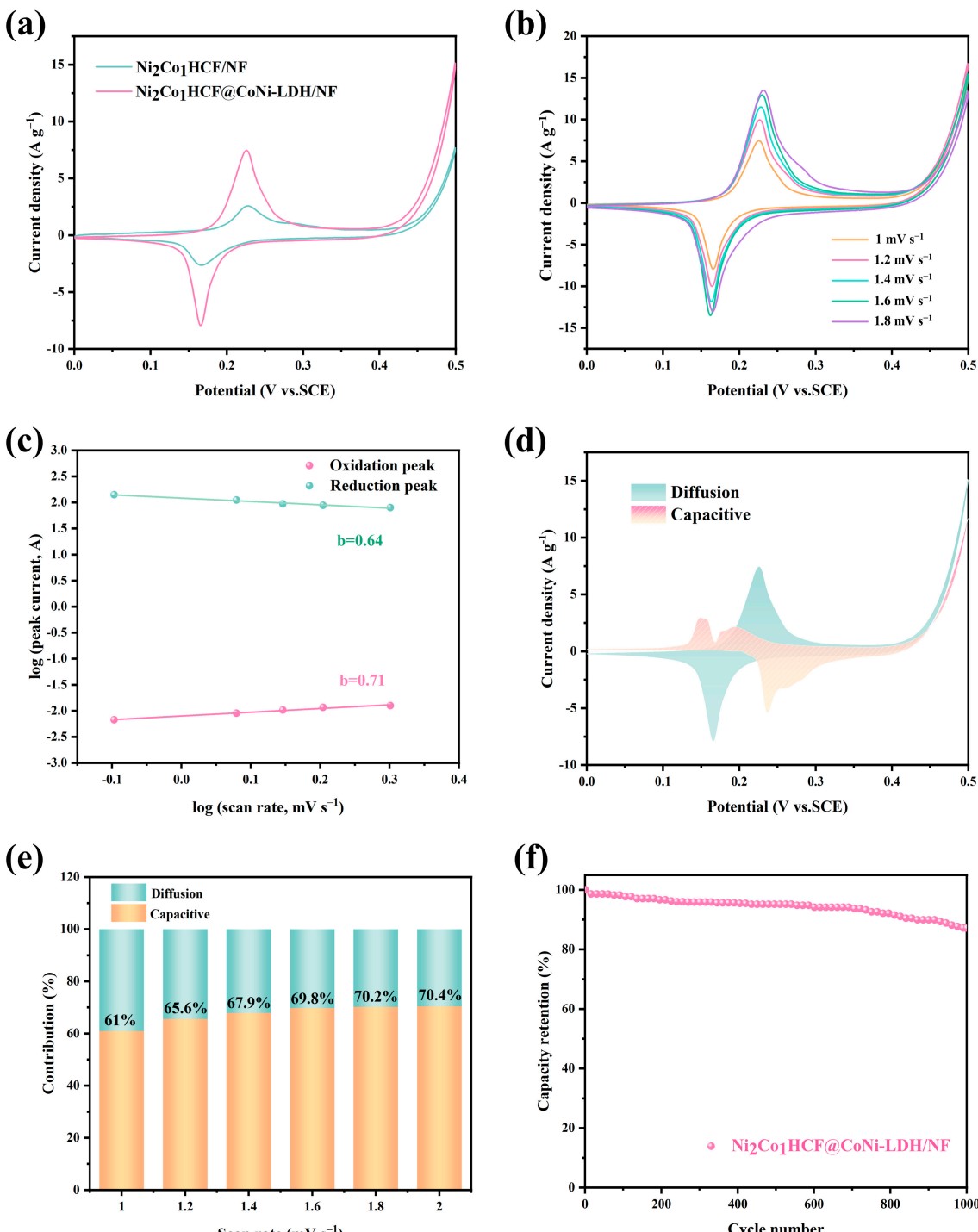

**Figure 6.** (**a**) Comparative CV curves of the $Ni_2Co_1HCF/NF$ and $Ni_2Co_1HCF@CoNi-LDH/NF$ composites; (**b**) CV curves at different scanning rates; (**c**) relationship between the logarithm of the peak cathode and anode currents and the logarithmic scan rate of the electrodes; (**d**) pseudocapacitance contribution versus diffusion contribution at a scan rate of 1 mV·s$^{-1}$; (**e**) contribution of pseudocapacitance at different scanning rates; (**f**) cycling stability curve at a current density of 5 A·g$^{-1}$ of the $Ni_2Co_1HCF@CoNi-LDH/NF$ composites.

## 4. Conclusions

In this work, $Ni_2Co_1HCF@CoNi-LDH$ composites were grown in situ on the surface of nickel foam by a combination of coprecipitation and constant potential electrodeposition. The composites maintained the inherent cubic skeleton morphology of the Prussian blue

material, and CoNi-LDH was successfully coated on the surface of the NiCo Prussian blue. The electrochemical performance of the composites were evaluated using a three-electrode system in 2 M KOH. The material had a high discharge specific capacity of 1937 $F \cdot g^{-1}$ at a current density of 1 $A \cdot g^{-1}$ and a capacity retention rate of 81.3%, even at a high current density of 10 $A \cdot g^{-1}$. Its GCD curve shows a clear plateau, which is typical of pseudocapacitance behavior, and its CV curve possessed a symmetrical and obvious pair of redox peaks, showing good reversibility. The electrochemical impedance spectra of the composites also showed smaller $R_{ct}$ values, indicating improved conductivity. In addition, the capacity retention rate reached 87.1% after 1000 cycles at a current density of 5 $A \cdot g^{-1}$, showing a good cycling stability. The calculation of the pseudocapacitance contribution and the diffusion kinetic analysis show that the charge storage process of the composite is controlled by both the ion diffusion process and pseudocapacitance behavior, but the pseudocapacitance behavior is dominant, and the pseudocapacitance contribution of the composite at 1 $mV \cdot s^{-1}$ reached 61%. Therefore, the $Ni_2Co_1HCF@CoNi$-LDH/NF composites have good application prospects as supercapacitor electrode materials and provide new ideas for the design of asymmetric supercapacitor electrode materials in the future.

**Author Contributions:** Conceptualization, Q.Y. and B.G.; methodology, Q.Y.; software, Q.Y.; validation, Q.Y., H.F. and L.H.; formal analysis, Q.Y.; investigation, Q.Y.; resources, H.F.; data curation, Q.Y.; writing—original draft preparation, Q.Y.; writing—review and editing, B.G.; visualization, Q.Y.; supervision, B.G.; project administration, B.G.; funding acquisition, B.G. All authors have read and agreed to the published version of the manuscript.

**Funding:** This research was supported by the National Natural Science Foundation of China (51671052), the Fundamental Research Funds for the Central Universities (N182502042, N2025001) and the Liao Ning Revitalization Talents Program (XLYC1902105).

**Institutional Review Board Statement:** Not applicable.

**Informed Consent Statement:** Not applicable.

**Data Availability Statement:** All data generated and analyzed during this study are included in this article.

**Conflicts of Interest:** The authors declare no conflict of interest.

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
