# Peer review of "NiCo Prussian-Blue-Derived Cobalt–Nickel-Layered Double Hydroxide with High Electrochemical Performance for Supercapacitor Electrodes"

_coatings, doi:10.3390/coatings13030554_

Round 1
Reviewer 1 Report
The manuscript is relatively well written and presented. The concept is interesting yet the experimental design is not perfectly executed. Therefore, this manuscript can be accepted.
Author Response
Thank you very much for your approval of this paper, we will further improve it according to the review comments.
Reviewer 2 Report
This study investigated the fabrication of Ni2Co1HCF@CoNi-LDH composites for supercapacitor application with a specific capacitance of 1937 F/g and retaining 87.1 % of the initial capacitance after 1000 cycles.
1. Introduction needs new references from Coatings journal
2. Equation needs a reference (1)….The specific capacitance of the supercapacitors using the cyclic voltammetry method can be obtained from the following reference: CNTs Supercapacitor Based on the PVDF/PVA Gel Electrolytes.
3. Schematic of the supercapacitor add to the manuscript
4. The Bode curves of the supercapacitor could be added to the manuscript
5. Schematic of an equivalent circuit of the supercapacitor in Nyquist curves should be added to the manuscript
6. Power density and energy density of supercapacitor should be add to the manuscript: such as the following references Design and fabrication of ultracapacitor based on paper substrate and BaTiO3/PEDOT: PSS separator film, Graphite nanoparticles paper supercapacitor based on gel electrolyte.
7. Add Power density and energy density to table 1
8. Conclusion should be improved
Author Response
Thank you very much for your review comments, the specific response has been put in a word file.

Reviewer 3 Report
In this manuscript, the composite Ni2CoHCF@CoNi-LDH with the core-shell structure was prepared by combining co-precipitation and constant potential electrodeposition. The structural and morphological properties were analyzed by XRD, SEM, FTIR, and XPS. Further, an application point of view, the electrochemical characteristics are measured. The results showed that its discharge-specific capacity was as high as 1937 F/g at a current density of 1 A/g and still had 81.3% capacity retention at 10 A/g, which is an exciting aspect of this work. The manuscript is suggested to be accepted after the following issues are addressed.
1) The authors measured the CV and GCD at different potential windows. Mostly, the devices were checked at the same window.
2) The authors should measure the CV of the electrode at the different potential windows to check the performance.
3) The authors measured the stability of the device up to 1000 cycles that one small application point of view.
4) The authors should also add XRD and SEM after measuring the stability of the electrode
5) Many spelling and formatting typos in this paper and the authors should improve them.
6) The authors should cite some latest work such as; 10.1016/j.electacta.2020.136039; 10.1038/s42004-019-0184-6
Author Response

(The authors gave the same response as above.)

Reviewer 4 Report
Manuscript can be published after major revisions based on the following comments:
1. Please define HCF, LDH and NF acronyms at the beginning of the manuscript;
2. Please, highlight the novelty of the manuscript with respect previous literature on materials for electrochemical capacitors;
3. Introduction section, line 51: “we propose an environmentally friendly”: the process you propose is not environmentally friendly since you use Co that is a toxic metal and [Fe(CN)6]3- that is considered as carcinogenic agent. Please, remove the sentence in Introduction section.
4. Reactions 6-8: what is the evidence of the formation of CoO2 and NiOOH after the operation of the electrodes? Please, provide proofs at support of the reaction mechanism you reported in the manuscript;
5. Figure 5d: please, report fitting lines in Figure and not lines connecting points.
6. Please, report a table with all the impedance spectra fitting parameters; without this table (with also chi2 values) it is really hard to comment impedance spectra results;
7. Lines 268-269: Rs is not 0.1 ohms in any case, this sentence is wrong. And, you need to explain why Rs is different by changing electrode material. Lines 269-271: I do not really understand the explanation about Rs, please clarify this point.
8. Why did you insert a Warburg element in the electrical equivalent circuit? What is the mass transfer-controlled process? Please, clarify in the text.
9. Line 295: “deformation” is not the right term. Please, amend the text
10. Manuscript title is too generic. You should mention in the title that the materials are synthesized for supercapacitor applications
Author Response
Response to Reviewer 4 Comments
Point 1: Please define HCF, LDH and NF acronyms at the beginning of the manuscript;
Response 1:The abbreviations HCF, LDH and NF have been defined at the beginning of the manuscript in accordance with the review comments.
Point 2: Please, highlight the novelty of the manuscript with respect previous literature on materials for electrochemical capacitors;
Response 2:We have highlighted the novelty in the manuscript, both in the comparison with other literature in Table 2 and in the description of the methods in the introduction.
Point 3: Introduction section, line 51: “we propose an environmentally friendly”: the process you propose is not environmentally friendly since you use Co that is a toxic metal and [Fe(CN)6]3- that is considered as carcinogenic agent. Please, remove the sentence in Introduction section.
Response 3:Sentences have been removed as requested by the reviewer, but we intended to express that it is more environmentally friendly compared to the hydrothermal method.
Point 4: Reactions 6-8: what is the evidence of the formation of CoO2 and NiOOH after the operation of the electrodes? Please, provide proofs at support of the reaction mechanism you reported in the manuscript;
Response 4:Since Ni in NiOOH is +3 valence, the presence of +3 valence Ni is evident from the Ni 2p fitting results in the XPS in our manuscript. For CoO2, since the material surface layer is a cobalt-nickel double hydroxide, we refer to similar literature “Enhanced ionic diffusion interface in hierarchical metal-organic framework@layered double hydroxide for high-performance hybrid supercapacitors”.
Point 5: Figure 5d: please, report fitting lines in Figure and not lines connecting points.
Response 5:The fitted curve is indeed shown in Figure 5(d).
Point 6: Please, report a table with all the impedance spectra fitting parameters; without this table (with also chi2 values) it is really hard to comment impedance spectra results;
Response 6:A table of impedance fitting results has been added as requested in the review, as shown in Table 1.
Point 7: Lines 268-269: Rs is not 0.1 ohms in any case, this sentence is wrong. And, you need to explain why Rs is different by changing electrode material. Lines 269-271: I do not really understand the explanation about Rs, please clarify this point.
Response 7:Thank you for pointing out my mistake, the value of Rs has been corrected. Rs in the manuscript can be understood as contact resistance, which includes electrolyte resistance and active substance internal resistance.
Point 8: Why did you insert a Warburg element in the electrical equivalent circuit? What is the mass transfer-controlled process? Please, clarify in the text.
Response 8:The impedance spectrum shows a clear electrochemical semi-infinite diffusion process (diagonal part), thus increasing the Warburg impedance. We have read the manuscript carefully and we did not find "the mass transfer-controlled process" in the manuscript. If you are referring to a semi-infinite diffusion process, we have added an explanation in the manuscript.
Point 9: Line 295: “deformation” is not the right term. Please, amend the text
Response 9:It has been revised according to the review comments, and “deformation” has been revised to “transform”.
Point 10: Manuscript title is too generic. You should mention in the title that the materials are synthesized for supercapacitor applications
Response 10:The title of the manuscript has been revised according to the review requirements, and the title has been changed to “NiCo Prussian Blue Derived Cobalt-Nickel Layered Double Hydroxide with High Electrochemical Performance for Supercapacitor Electrodes”
Reviewer 5 Report
The paper report the development of a new pseudocapacitor based on the modification of a Prussian blue like structure by using Co and Ni. The material was fabricated in a two-step method combining a classical chemical route together a later electrodeposition. The resulting electrode material was correctly morphological and structural characterized by several techniques (e.g.: EDX,FTIR,TEM, etc.). Concerning the improvements obtained, the material possesses higher capacity retention than other similar ones found in literature.
The paper is consistent and well addressed also, in my opinion, the scope of the journal is suitable for this work. However, there are several minor issues that must be addressed before publication. That is why I recommend publication in the Coatings journal after some minor revisions.
Specific remarks:
1) In the abstract it is recommended remove the following sentence ” The electrochemical properties of the composites were investigated using a three-13 electrode system in 2 M KOH” this information is more suitable to be presented in the experimental section. Besides, this extra space can be filled with more relevant information.
2) The origin of the reagents should be stated as well at least the country of the company (lines 67,69, etc.)
3) More details concerning the manner to sonicate the solution in line 78 should be included.
4) The writing of section 2.3. should be revised, it is kind of wordy and information is repeated through it.
5) More information about SEM assays should be included in lines 101-102.
6) It is recommended to include a reference concerning the Equation 1 (line120).
Minor remarks
1) In the abstract it is recommended to include the potential employed for the electrodeposition (line 8).
2) If possible, place the reference number at the end of the sentence to disturb less the reader (e.g.: line 44 “[7]” can be placed in line 47 before “Xiong et al”.
3) In lines 143-144 the size of “(3)” and “(4)” is different from “(2)”.
4) In line 225 the heading is incorrect, I assume it should be “3.2 Electrochemical performance".
5) Figure 6e has wrong labels, now it reads “capqacitive”.
Author Response
Response to Reviewer 5 Comments
Specific remarks:
Point 1: In the abstract it is recommended remove the following sentence ” The electrochemical properties of the composites were investigated using a three-13 electrode system in 2 M KOH” this information is more suitable to be presented in the experimental section. Besides, this extra space can be filled with more relevant information.
Response 1:We are very grateful for the reviewer's suggestion and we think that the abstract "The electrochemical properties of the composites were investigated using a three electrode system in 2 M KOH" is necessary to clarify directly the reaction environment and the test system, i.e., the method used for electrochemical testing.
Point 2: The origin of the reagents should be stated as well at least the country of the company (lines 67,69, etc.)
Response 2:The country where the reagent source company is located has been added to the manuscript.
Point 3: More details concerning the manner to sonicate the solution in line 78 should be included.
Response 3:The description of the ultrasonic treatment is not detailed in most research papers, because the ultrasonic treatment in this manuscript is performed using ultrasonic cleaning machines as in other research papers, which can be found in the literature 10.1016/j.jpowsour.2020.227712 and 10.1016/j.carbon.2021.12.097.
Point 4: The writing of section 2.3. should be revised, it is kind of wordy and information is repeated through it.
Response 4:Thank you very much for the reviewer's suggestion. The contents of parts 2 and 3 have been revised, and the second part mainly describes the experimental steps, and the third part mainly describes the experimental principles.
Point 5: More information about SEM assays should be included in lines 101-102.
Response 5:A description of the SEM analysis has been added.
Point 6: It is recommended to include a reference concerning the Equation 1 (line120).
Response 6:References on equation (1) have been added.
Minor remarks
Point 1: In the abstract it is recommended to include the potential employed for the electrodeposition (line 8).
Response 1:We thank the reviewers for their suggestions, and we think it would be better to mention the deposition potential in the experimental method, because it may be duplicated in the abstract, and in addition, the abstract focuses on the experimental method.
Point 2: If possible, place the reference number at the end of the sentence to disturb less the reader (e.g.: line 44 “[7]” can be placed in line 47 before “Xiong et al”.
Response 2:The reference number has been placed at the end of the sentence in accordance with the review comments.
Point 3: In lines 143-144 the size of “(3)” and “(4)” is different from “(2)”.
Response 3:The font size has been corrected in accordance with the review requirements.
Point 4: In line 225 the heading is incorrect, I assume it should be “3.2 Electrochemical performance".
Response 4:The title number has been revised according to the review requirements.
Point 5: Figure 6e has wrong labels, now it reads “capqacitive”.
Response 5:The spelling errors in the images have been corrected in accordance with the review requirements.
Round 2
Reviewer 2 Report
Dear Editor
The manuscript well has been revised
Reviewer 3 Report
Accept in present form.
Reviewer 4 Report
The manuscript is now suitable for publication.